# Integer-Only CNNs with 4 Bit Weights and Bit-Shift Quantization Scales at Full-Precision Accuracy

**Maarten Vandersteegen** * [ID] **, Kristof Van Beeck** [ID] **and Toon Goedemé** [ID]

KU Leuven, EAVISE-Jan Pieter De Nayerlaan 5, 2860 Sint-Katelijne-Waver, Belgium;
kristof.vanbeeck@kuleuven.be (K.V.B.); toon.goedeme@kuleuven.be (T.G.)
* Correspondence: maarten.vandersteegen@kuleuven.be; Tel.: +32-15-31-69-44

**Abstract:** Quantization of neural networks has been one of the most popular techniques to compress models for embedded (IoT) hardware platforms with highly constrained latency, storage, memory-bandwidth, and energy specifications. Limiting the number of bits per weight and activation has been the main focus in the literature. To avoid major degradation of accuracy, common quantization methods introduce additional scale factors to adapt the quantized values to the diverse data ranges, present in full-precision (floating-point) neural networks. These scales are usually kept in high precision, requiring the target compute engine to support a few high-precision multiplications, which is not desirable due to the larger hardware cost. Little effort has yet been invested in trying to avoid high-precision multipliers altogether, especially in combination with 4 bit weights. This work proposes a new quantization scheme, based on power-of-two quantization scales, that works on-par compared to uniform per-channel quantization with full-precision 32 bit quantization scales when using only 4 bit weights. This is done through the addition of a low-precision lookup-table that translates stored 4 bit weights into nonuniformly distributed 8 bit weights for internal computation. All our quantized ImageNet CNNs achieved or even exceeded the Top-1 accuracy of their full-precision counterparts, with ResNet18 exceeding its full-precision model by 0.35%. Our MobileNetV2 model achieved state-of-the-art performance with only a slight drop in accuracy of 0.51%.

**Keywords:** quantization; neural networks; nonuniform; power-of-two scales; low-cost hardware

## 1. Introduction

Quantization of neural networks dates back to the 1990s [1,2], where the discretization of models was a necessity to make their implementation feasible on the available hardware. More recently, neural networks became popular again because of the ImageNet challenge [3] and the availability of powerful GPU hardware. This breakthrough started a new area of research with hundreds of new potential applications. Today, neural networks are found in various electronic devices such as smartphones, wearables, robots, self-driving cars, smart sensors, and many others. The embedded electronic processors found in these applications are often very limited in capabilities to make them affordable, compact, and energy efficient in the case of battery-powered applications. Most neural networks are too large and require too many computations to be implemented directly on such processors, and therefore need to be compressed first. One of the most effective ways to reduce latency, storage cost, memory-bandwidth, energy efficiency, and silicon area among popular compression techniques such as model pruning [4] and network architecture search [5] is model quantization [6]. The quantization of neural networks is a frequently visited research topic with numerous publications that mostly focus on reducing the number of bits per weight or activation as much as possible in order to achieve high compression rates [7–11].

The commonly accepted method to achieve low-precision quantization introduces high-precision scale factors, and in some cases zero-points, to adapt the quantized values to the diverse weight and activation ranges present in full-precision neural networks. This

is done to avoid major degradation in accuracy. The downside of this approach is that these high-precision quantization scales have to be computed at runtime as well in order to make the math work. Because these high-precision multipliers handle the minority of computations, their involvement is not considered a problem by most works. Their required presence in hardware, however, increases silicon area, energy consumption, and cost. In addition, neural network compute engines implemented on FPGAs may require more expensive FPGAs, due to the limited number or lack of high-precision multipliers on less expensive platforms. This is especially problematic when multiple compute cores are used to increase parallelism.

Jain et al. [12] already proved that 8 bit models with per-layer quantization and power-of-two scales can achieve full-precision accuracy or even better. Their experiments with 4 bit weights and 8 bit activations, however, resulted in a significant degradation in accuracy and even a complete failure for MobileNets. Using 4 bit weights, 8 bit activations, and per-layer quantization with power-of-two scales, we prove that we can achieve full-precision accuracy or even better for all models and near full-precision accuracy for MobileNetV2. Moreover, we prove that our method performs on-par or even better compared to uniformly per-channel quantized models that use full-precision quantization scales. Our proposed compute engine is depicted in Figure 1b, next to a typical compute engine [6,13,14] with high-precision scaling capability in Figure 1a. We propose to use a lookup-table to translate 4 bit weight addresses into 8 bit weights for internal computation. Since the LUT can hold any number representable through 8 bit, different value combinations can be chosen to match the underlying weight distribution within a specific convolution layer, compensating the limited capabilities of the bit-shift scaling. A different set of LUT values is used for every layer to best match the layer-specific data distribution. Note that a single LUT can be shared among multiple parallel compute engines, allowing additional hardware simplification.

Reducing the number of bits per weight from eight to four is a wanted feature in many cases because it reduces the storage cost by 50% and decreases the load-time of the weights from external memory significantly, which results in faster computation. With our method, 32 bit (integer) multipliers can be avoided, transforming the neural network compute engine to simple, yet very effective hardware.

Our contributions can be summarized as follows:

- We present an extensive literature overview of uniform and nonuniform quantization for fixed-point inference;
- A novel modification to a neural network compute engine is introduced to improve the accuracy of models with 4 bit weights and 8 bit activations, in conjunction with bit-shift-based scaling, through the aid of a lookup-table;
- A quantization-aware training method is proposed to optimize the models that need to run on our proposed compute engine;
- We are the first to make a fair empirical comparison between the performance of (uniform) quantized models with full-precision and power-of-two scales with either per-layer or per-channel quantization using 4 bit weights;
- Our source code has been made publicly available https://gitlab.com/EAVISE/lut-model-quantization (accessed on 16 November 2021).

The remainder of this paper is organized as follows: Section 2 presents an extensive literature overview of quantization in greater detail, organized into different topics for convenience. For each topic, we also highlight the choices we made for our own approach. Our proposed method is explained in Section 3; our results are presented in Section 4; conclusions are made in final Section 5.

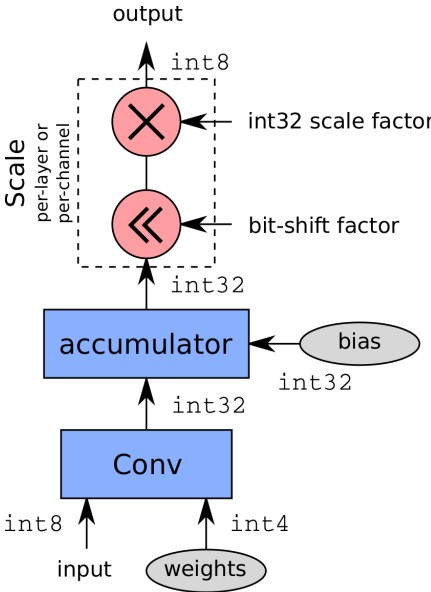

(**a**) Typical

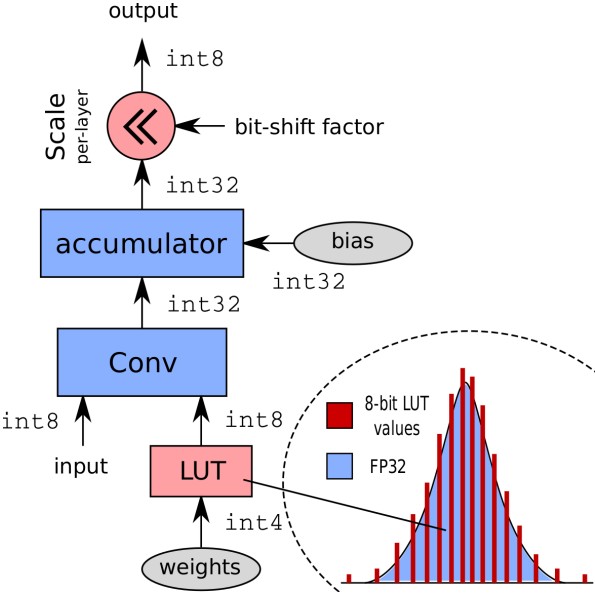

(**b**) Ours

**Figure 1.** (**a**) Typical fixed-point compute engine of a convolutional layer; (**b**) our proposed fixed-point compute engine with a single per-layer power-of-two scale (bit-shift) and a lookup-table to translate 4 bit weights from storage to 8 bit weights for internal computation. The LUT is used to boost the expressive power of the engine through nonuniform weight quantization, which compensates the lack of high-precision scaling.

## 2. Related Work

Related work on neural network quantization algorithms can be divided in two major categories: Post-Training Quantization (PTQ) and Quantization-Aware Training (QAT). Post-training quantization algorithms quantize the weights and activations without retraining the model. This is the fastest approach and usually results in near-floating-point performance for 8 bit weights and activations and requires no training data during quantization. Quantization-aware training is the way to resort to when the accuracy drop between a PTQ-quantized model and its baseline is too large, which is for example the case

when bit-widths lower than eight are used. In this literature review, QAT methods are the main focus since our method belongs to this category.

This literature review is structured into several topics for clarity. First, the basic principles of quantization and their impact on the target compute engine are introduced in Section 2.1. Section 2.2 elaborates on modeling quantization behavior in order to simulate the target hardware, and Section 2.3 discusses common techniques to relax the quantization constraints (e.g., symmetric versus asymmetric quantization, per-layer versus per-channel quantization, and uniform versus nonuniform quantization). Section 2.4 elaborates on data types and different estimation techniques for the quantizer parameters. Section 2.5 describes the quantization of nonparametric layers such pooling layers, elementwise adders and concatenation, and finally, the importance of the initialization of QAT is highlighted in Section 2.6.

### 2.1. Quantization Principles

The basic principles of quantization in neural networks and their impact on the compute engine are introduced in this section using the most simple, but widely used uniform symmetric quantization scheme. In Section 2.3, other more complex schemes are discussed as well.

#### 2.1.1. Uniform Symmetric Quantization

In a uniform symmetric quantization scheme, a floating-point vector or tensor $\mathbf{x}$ within a quantized neural network is approximated by a signed b bit integer vector $\mathbf{q}$ and a high-precision scale factor $s$:

$$\mathbf{x} \approx s\mathbf{q} \tag{1}$$

It is called symmetric because the number of discrete values is symmetrically distributed around zero. Scale factor $s$ is used to regulate the distance between the discrete steps of $\mathbf{q}$. A small value of $s$ results in smaller steps and, thus, higher precision, but decreases the minimum and maximum value that can be represented with b bits. A higher value of $s$ results in a higher minimum and maximum, but sacrifices precision in turn. Using this scheme, a trade-off between range and precision can be selected that is suited for a specific part within a neural network. Usually, each layer obtains its own scale factor since differences in the distribution range between layers can be large.

The bulk of the computations in the convolution and fully connected layers is performed by a multiply-accumulate engine, given by Equation (2).

$$y = o + \sum^{N} \mathbf{w} \odot \mathbf{a} \tag{2}$$

This engine calculates an output value $y$ from a weight vector $\mathbf{w}$, activation vector $\mathbf{a}$, and bias or offset value $o$. A typical quantized or integer version of Equation (2) can be created by replacing all floating-point values with the quantization scheme from Equation (1):

$$(s_y y_q) = (s_o o_q) + \sum^{N} (s_w \mathbf{w}_q) \odot (s_a \mathbf{a}_q) \tag{3}$$

$$y_q = \frac{s_o o_q}{s_y} + \frac{s_w s_a}{s_y} \sum^{N} \mathbf{w}_q \odot \mathbf{a}_q \tag{4}$$

Separate scales $s_w$, $s_a$, and $s_o$ are assigned to integer weights $\mathbf{w}_q$, integer activations $\mathbf{a}_q$, and integer bias $o_q$, respectively, to give more flexibility to the quantization process. In practice, the value of $s_o$ is not that critical since the number of bits for the bias is typically high (32 bit signed integer). Jacob et al. [13] therefore proposed to set $s_o = s_w s_a$, which further simplifies the integer kernel to:

$$y_q = S\left(o_q + \sum_{}^{N} \mathbf{w}_q \odot \mathbf{a}_q\right) \quad s.t. \quad S = \frac{s_w s_a}{s_y} \tag{5}$$

A typical example would be 8 bit weights and activations and a 32 bit bias. In this case, the engine would accumulate the multiplication result in a wide accumulator of typically 32 bit and would add the bias to the result. Then, a *requantization* step is applied, which involves the following consecutive operations: a single high-precision scale operation with scale factor $S$, a round operation (optional), a clipping operation that saturates the result, typically in the range $[-128, 127]$, and finally, a type conversion that casts the result back to 8 bit. Results from the compute engine are written back to memory in order to be used as the input for the next layer. Besides the high-precision scale operation with $S$, the whole engine would only require 8 bit multipliers and 32 bit accumulators. Section 2.4.1 discusses in more detail how $S$ is typically represented in hardware.

### 2.1.2. Activation Functions

In neural network graphs, nonlinearities are often placed directly after a linear layer, such as a convolution or fully connected layer. It would be wasteful if the linear layer would write its result back to memory, to later fetch it again to apply the nonlinearity. Many compute engines are therefore equipped with a unit that can execute a nonlinearity before applying the requantization step. The requantization step can in fact already model functions such as the ReLU or ReLU6, by simply setting the lower clipping bound to zero and, in the special case of ReLU6, lowering the upper bound as well. These functions can therefore be executed with virtually no overhead. Figure 2a gives an overview of a fixed-point convolutional engine with built-in activation function support. Moving the activation function into the compute engine is also referred to as the *fusion* of the activation function. In case an activation function is not supported by the compute engine, it needs to be executed separately, resulting in additional load/storage overhead.

### 2.1.3. Folding Batch Norm Layers

Batch normalization layers [15] are very popular in convolutional networks because they drastically speedup the training process and push the model to higher accuracies. During inference, however, they are nothing more than additional overhead, and therefore, their parameters are usually *folded* into the preceding convolution layer as an optimization step, prior to quantization. The folding process simply rescales the weights of the convolution and modifies its bias as given by Equations (6) and (7).

$$\mathbf{w}_{fold} = \frac{\gamma \mathbf{w}}{\sqrt{\sigma^2 + \epsilon}} \tag{6}$$

$$o_{fold} = \frac{(o - \mu)}{\sqrt{\sigma^2 + \epsilon}} \gamma + \beta \tag{7}$$

$\mu$, $\sigma$, $\mathbf{w}$, and $o$ are the running mean, running standard deviation, original weights, and bias, respectively. $\gamma$ and $\beta$ represent the trained parameters of the batch normalization layer.

### 2.2. Simulating Quantization

To test what the accuracy of a quantized neural network would be on a fixed-point processing device, quantization frameworks can simulate the behavior of the device in a floating-point environment. As such, there is no need to port the model to the physical device each time a test needs to be conducted. A simulation of a quantized device is also required with QAT: it allows a training framework to simulate the quantization behavior during training to allow the weights to adapt to the discrete sample space.

The common approach for simulating a quantized device is by inserting so-called *fake* quantization nodes throughout the full-precision network graph. These nodes are

called *fake* because they simulate the quantization behavior of the target hardware using full-precision floating-point values rather than actually quantizing the model.

Figure 2b gives a typical simulated version of the compute engine from Figure 2a, where $Q_f$ represents a *fake* quantization node. The precision of 32 bit integer operations is usually comparable to that of floating-points, which is why *fake* quantization nodes are omitted at the bias-add operation in Figure 2b, to save memory and computational overhead during training. If lower precision accumulators are used (e.g., 16 bit or less), the simulated scheme in Figure 2c is preferred. Note that a *fake* quantization node *before* the activation function is also omitted because this simulation assumes that the activation function will be fused into the preceding convolution on the target device. If the activation function needs to be executed separately, an additional $Q_f$ is likely needed before the activation function.

Section 2.2.1 discusses the internals of a *fake* quantizer.

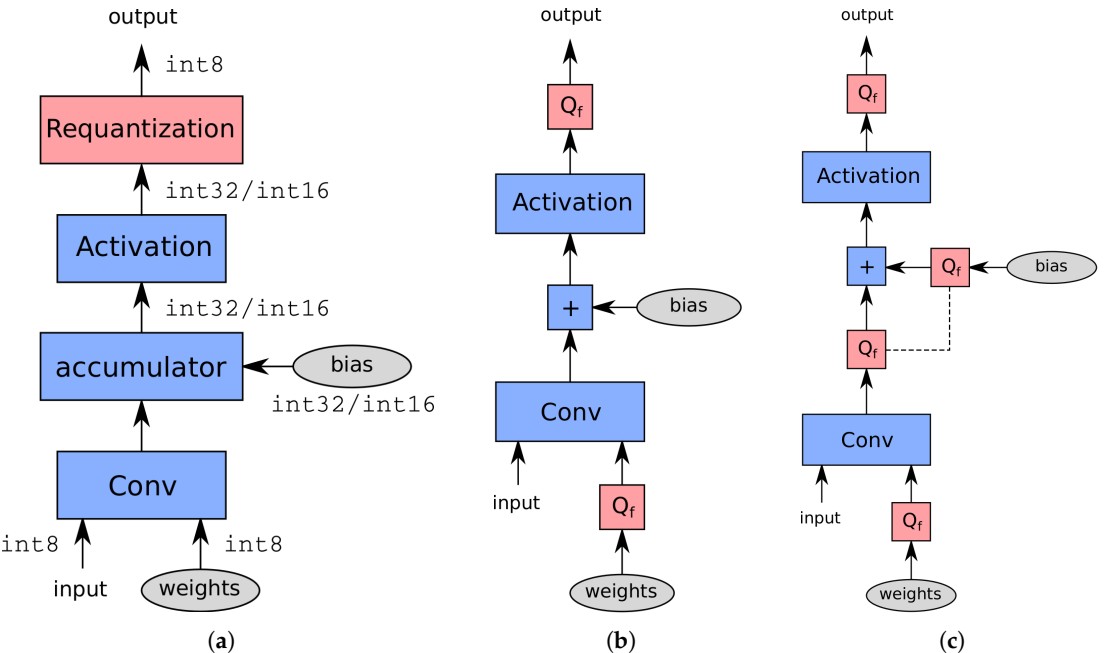

**Figure 2.** (**a**) Fixed-point compute engine of a convolutional layer; (**b**) simulated fixed-point compute engine with 32 bit accumulator and bias; (**c**) simulated compute engine with 16 bit (or less) accumulator and bias. The dashed line indicates that these *fake* quantization nodes share the same scale factor.

### 2.2.1. *Fake* Quantizer

A *fake* quantizer or quantization node has a quantization step and a dequantization step:

$$Q_f(\mathbf{x}) = D(Q(\mathbf{x})) \tag{8}$$

$Q$ is a quantization operator that will project the floating point values in $\mathbf{x}$ to discrete values. $D$ is a dequantization operator that will revert the quantized values back to floating-point values within the same value range as input $\mathbf{x}$.

Equations (9) and (10) give a typical example of a uniform symmetric quantization and dequantization function, respectively, which simulate the quantization scheme from Equation (1). Equation (11) combines both equations together.

$$Q(\mathbf{x}, s, b) = \text{clamp}\left(\text{round}\left(\frac{\mathbf{x}}{s}\right), -2^{b-1}, 2^{b-1} - 1\right) \tag{9}$$

$$D(\mathbf{q}) = s\mathbf{q} \tag{10}$$

$$Q_f(\mathbf{x}, s, b) = s\,\text{clamp}\left(\text{round}\left(\frac{\mathbf{x}}{s}\right), -2^{b-1}, 2^{b-1} - 1\right) \tag{11}$$

The *round* operator rounds input values to integer values, and *clamp* saturates the data to simulate the upper and lower limits of a signed fixed-point representation. The number of bits is defined by $b$, and $s$ is the scale factor. The dequantization operation simply multiplies the quantized integer values with the scale factor. If for example $b = 8$ and $\mathbf{x} \in [-10, 10]$, then it would make sense to set $s = \frac{10}{128}$. $Q_f$ will be:

$$Q_f\left(\mathbf{x}, s = \frac{10}{128}, b = 8\right) = \frac{10}{128} \, \text{clamp}\left(\text{round}\left(\frac{128}{10} \cdot \mathbf{x}\right), -128, 127\right) \tag{12}$$

The output of $Q_f(\mathbf{x}, s = \frac{10}{128}, b = 8)$ will be floating-point numbers, but their values are restricted to 256 discrete levels symmetrically distributed around 0. Values outside $[-10, 10)$ are saturated within $[-10, 10)$ through the *clamp* operator. Scale factor $s$ is a quantizer parameter that needs to be determined as a function of $\mathbf{x}$. Its value needs to be chosen carefully since it will determine the balance between precision and range. Proper selection of $s$ is discussed in Section 2.4.2.

Equation (11) describes a *fake* quantizer in the forward pass. If the simulation of quantization is needed during QAT, $Q_f$ must be able to propagate gradients during the backward pass. The local gradient of a *round* function is, however, zero almost everywhere, which makes gradient descent based on *round* operators and, hence, on *fake* quantization nodes impossible. The common solution to this problem is the *Straight Through Estimator* (STE) [16] principle, which simply bypasses the *round* operator in the backward pass, treating it as an identity function:

$$\frac{\partial \, \text{round}(x)}{\partial x} = 1 \tag{13}$$

Figure 3a,b illustrates the transfer function of an identity function and a *round* function, respectively, with their corresponding derivatives, while Figure 3c depicts a *round* function using the STE principle. Using the STE, the local gradient of the *fake* quantizer (Equation (11)) w.r.t. input $\mathbf{x}$ becomes:

$$\frac{\partial Q_f(\mathbf{x}, s, b)}{\partial \mathbf{x}} = \begin{cases} 1 & \text{if} - 2^{b-1} \leq \text{round}\left(\frac{\mathbf{x}}{s}\right) \leq 2^{b-1} - 1 \\ 0 & \text{otherwise} \end{cases} \tag{14}$$

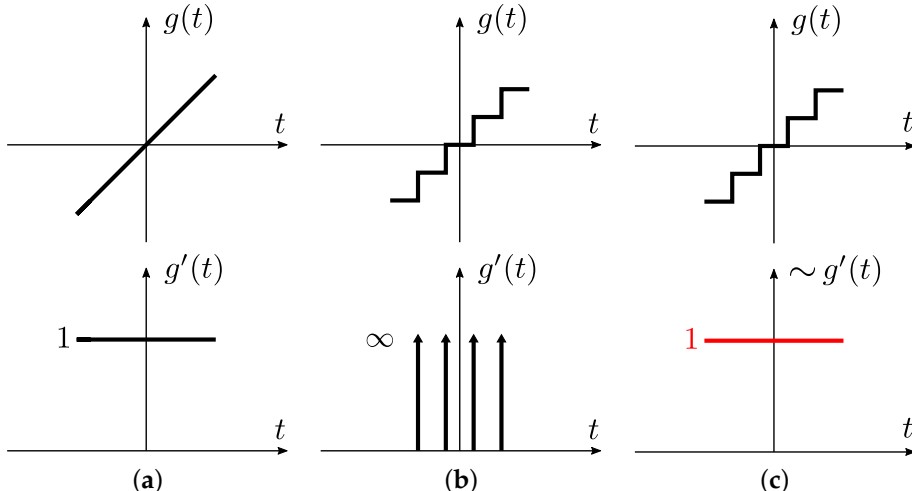

**Figure 3.** At the top, transfer functions of an identity function (**a**) and a *round* function (**b**) with their respective derivatives at the bottom. A round function with the STE in the backward pass is depicted in (**c**).

Some works argue that the STE produces suboptimal results because it uses an estimate rather than the true gradient. An alternative solution proposes to approximate the round

operator with a linear combination of several sigmoid functions to obtain a staircase-like transfer function [17]. Another work [18] proposed *alpha blending*, a technique that linearly combines the original data $\mathbf{x}$ with quantized data $Q_f(\mathbf{x}, s, b)$ through a blending factor $\alpha$ in the forward pass:

$$Q_{final}(\mathbf{x}, s, b, \alpha) = \alpha Q_f(\mathbf{x}, s, b) + (1 - \alpha)\mathbf{x} \tag{15}$$

$Q_f(\mathbf{x}, s, b)$ blocks the gradient in the backward pass, while term $(1 - \alpha)\mathbf{x}$ propagates it, which results in *true* gradient calculation. During training, $\alpha$ is gradually increased from 0 to 1. This alternative solution, however, has limited added value with respect to accuracy. In addition, the constantly increasing $\alpha$ makes it impractical to use quantizer scales that are learned through backpropagation, which rules out the modern way of estimating the scale factor with QAT. We therefore chose to use the well-proven STE within our work instead.

### 2.2.2. Folding Batch Normalization Layers and Simulation

As mentioned in Section 2.1.3, batch normalization layers are folded prior to quantization, which is commonly called *static* folding. With QAT, however, other options exist. Krishnamoorthi et al. [14] argued that the effect of batch normalization is still useful during the QAT process, but at the same time, they wanted to avoid them during inference on the target. They solved this by applying the *folding* process (Equations (6) and (7)) before each forward pass, but using the training batch statistics $\mu_B$ and $\sigma_B$ to compute the folded weights and bias rather than the running statistics $\mu$ and $\sigma$. In this way, they preserved the batch normalization effect during QAT and, at the same time, simulated the folding process to prepare the model for future folding. Halfway during the QAT process, batch normalization layers are folded using the running statistics $\mu$ and $\sigma$. To calculate $\mu_B$ and $\sigma_B$, however, each preceding convolution must be forwarded a second time, which generates additional compute overhead during training. Some works [6,19] argued that fine-tuning with batch statistics is unnecessary and even harmful to the final accuracy with 4 bit weights. We confirmed the same behavior and found that applying the *folding* process with running statistics in every forward pass, as suggested by Li et al. [19], works slightly better compared to permanent folding prior to training, as proposed by Nagel et al. [6]. We therefore adopted the proposal from Li et al. [19] for all our experiments.

### 2.3. Relaxation of Quantization Constraints

In addition to the uniform symmetric quantizer, more complex quantization schemes have also been proposed in literature. Many of these schemes have a direct impact on the design of the compute engine and, therefore, require a good understanding. In this section, we discuss popular schemes including the uniform asymmetric quantization in Section 2.3.1, the per-channel quantization in Section 2.3.2, various nonuniform quantizations in Section 2.3.3, and mixed precision in Section 2.3.4.

### 2.3.1. Asymmetric Quantization

Equation (1) describes a so-called symmetric quantization scheme since it distributes the quantization levels equally around zero, assuming $\mathbf{q}$ has a signed integer data type. In case the input data are not symmetrically distributed around zero, a significant portion of the quantization bins might be unused when relying on a symmetric scheme. Therefore, many works prefer an asymmetric scheme, which adds a zero-point $z$ to the equation:

$$\mathbf{x} \approx s(\mathbf{q} - z) \tag{16}$$

In practice, both $\mathbf{q}$ and $z$ are unsigned integer types since $z$ can nudge the distribution to the desired place. A *fake* quantizer that supports this scheme usually becomes:

$$Q_f(\mathbf{x}, s, z, b) = s\left[\text{clamp}\left(\text{round}\left(\frac{\mathbf{x}}{s}\right) + z, 0, 2^b - 1\right) - z\right] \tag{17}$$

Although the asymmetric design offers superior properties compared to the symmetric design [6,14], it comes with an additional cost at inference time when both weights and activations use this scheme. Equation (21) shows the computations such a multiply-accumulate engine needs to make. Note that the zero-point for the bias is usually omitted [13]:

$$s_y(y_q - z_y) = s_o o_q + \sum^N s_w(\mathbf{w}_q - z_w) \odot s_a(\mathbf{a}_q - z_a) \tag{18}$$

$$y_q - z_y = S\left(o_q + \sum^N (\mathbf{w}_q - z_w) \odot (\mathbf{a}_q - z_a)\right) \quad s.t. \quad S = \frac{s_w s_a}{s_y}; \quad s_o = s_w s_a \tag{19}$$

$$y_q = z_y + S\left(o_q + \sum^N \mathbf{w}_q \odot \mathbf{a}_q - z_a \sum^N \mathbf{w}_q - z_w \sum^N \mathbf{a}_q + N z_w z_a\right) \tag{20}$$

$$y_q = S\left(O + \sum^N \mathbf{w}_q \odot \mathbf{a}_q - z_w \sum^N \mathbf{a}_q\right) \quad s.t. \quad O = \frac{z_y}{S} + o_q - z_a \sum^N \mathbf{w}_q + N z_w z_a \tag{21}$$

Since a few terms in Equation (20) are constant, these terms can be absorbed within a new bias $O$. Compared to Equation (5), the asymmetric engine Equation (21) needs to compute an additional term $-z_w \sum^N \mathbf{a}_q$ since it relies on runtime data. Although the computational cost does not increase much, it introduces additional complexity within the accelerator engine. For this reason, recent works [6] and deployment tools such as TensorFlow Lite use asymmetric quantization for their activations and symmetric quantization for the weights, which eliminates the calculation of the additional term at runtime.

Asymmetric quantizers support any given zero shift in their underlying data distribution in an optimal way. However, a symmetric quantizer with an unsigned integer data type could also be used in case the data solely contain positive values, which is for example the case with the ReLU or ReLU6 activation data. The quantization grid can then be optimally distributed without the need for an asymmetric quantizer. A symmetric *fake* quantizer for unsigned data could be defined as:

$$Q_f(\mathbf{x}, s, b) = s \, \text{clamp}\left(\text{round}\left(\frac{\mathbf{x}}{s}\right), 0, 2^b - 1\right) \tag{22}$$

Our proposed method solely uses symmetric quantizers for the activations with signed and unsigned data types since this works sufficiently well for 8 bit and eliminates additional complexity in estimating the zero-points.

### 2.3.2. Per-Channel Quantization

Until now, we only considered a single scale factor (and zero-point) for a whole weight or feature-map tensor. This is commonly referred to as *per-layer*, *per-axis*, or *scalar* quantization. Many works [6,14,20,21] have also experimented with separate quantizer parameters for each tensor channel, which is referred to as *per-channel* quantization. As asymmetric quantization is to symmetric quantization, *per-channel* quantization is another relaxation technique to better adapt to the different data distributions found in full-precision neural networks. This is particularly useful when the dynamic range within a single data tensor is large, which is often the case for networks with depthwise convolutions (e.g., MobileNets). Goncharenko et al. [21] reported a large difference in *per-layer* (scalar mode) and *per-channel* mode (vector mode) for MobileNetv2 and MNasNet. In practice, per-channel quantization is only applied for weights and not for activations because it would otherwise make the compute engine impractical to implement [6,14].

Nonetheless, per-channel quantization for the weights requires special considerations in hardware: a separate scale and optional zero-point need to be handled for each individual channel, which is not supported by all hardware accelerators. For this reason, Nagel et al. [22] proposed Cross-Layer Equalization (CLE), a per-channel weight-rescaling technique that is applied on the floating-point model, prior to quantization. CLE equalizes the weight tensors across their channel dimensions, achieving per-channel quantization

performance with PTQ. Nagel et al. [6] later discovered that CLE can also be used to initialize QAT, leading to better results.

Since we stress the simplicity of the hardware in this article, we chose to use per-layer quantization in our proposed method in conjunction with cross-layer equalization for models sensitive to quantization such as MobileNets.

### 2.3.3. Nonuniform Quantization

Linear or uniform refers to evenly distributed quantization levels among a given range. Uniform quantization schemes are usually preferred because of their simplicity and straightforward hardware implementation. Weight and activation data, however, tend to be nonuniformly distributed, as depicted in Figure 4. Several articles [10,17,20,23–25] proposed using quantization schemes with nonuniformly distributed quantization levels that more or less follow the expected distributions. Because weights tend to follow bell-shaped distributions, a more efficient precision/range balance can be achieved if more quantization levels are located close to the center and less levels are located further outwards, which results in higher accuracy. Proposed techniques in the literature include 1D K-means clustering of the weight/activation data, where the K cluster centers represent the quantization levels [20,26], cumulative histograms to create a mapping function that distributes the weights [20], and many more.

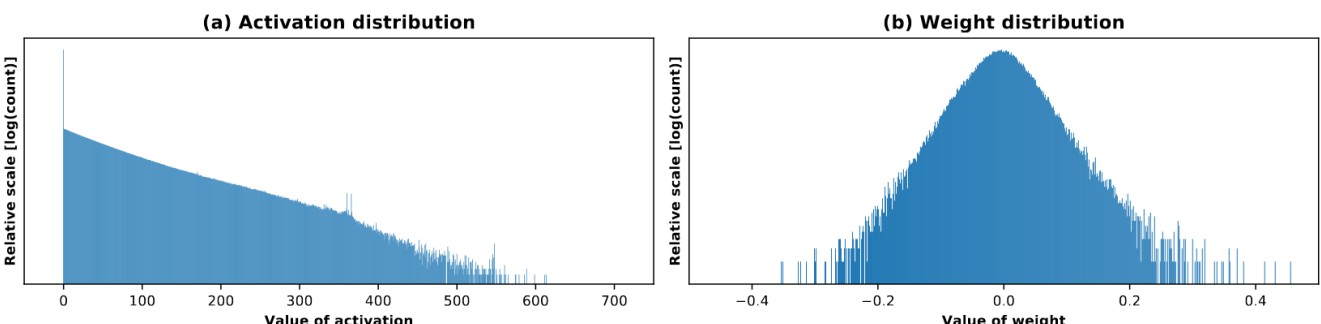

**Figure 4.** Activation and weight distribution of the second layer of GoogleNet (image from [27]).

Some works chose nonuniform schemes in a way that they can be realized very efficiently in hardware. One of the most popular schemes is the powers-of-two scheme, since this scheme has the potential to replace all multiplications with shift operations, which are faster and more energy efficient. Zhou et al. [25] proposed a training scheme where they incrementally replaced floating-point weights with powers-of-two weights during the QAT procedure. Li et al. [24] proposed to use a sum of two powers-of-two terms, which has some advantages over regular power-of-two distributions. Zang et al. [10] proposed LQ-Nets with a convolution engine that merely relies on bitwise operators, such as *xnor* and *popcount*, which can be executed very efficiently. Their engine supports a nonuniform quantization level distribution that can be trained and has a similar degree of freedom as the sum of power-of-two values. Faraone et al. [23] proposed to use a *Reconfigurable Constant Coefficient Multiplier* (RCCM) instead of bulky DSP multipliers in their FPGA implementation. RCCMs multiply numbers merely by shift and addition operations. They proposed a method to estimate the RCCM design based on an RCCM supported coefficient set that matches the distribution of the data as close as possible.

Covel et al. [20] proposed a lookup-table-based compute engine where all multiplications are omitted by implementing a double-indexed table that stores the multiplications results, calculated offline, between weights and activations. An additional lookup-table was used to translate the accumulated lookup results back to an activation index as the input for the next layer. Although very efficient, large tables are needed, which require a significant loading time between every network layer.

Our method was inspired by the additional accuracy gain that could be achieved through nonuniform quantization, since we relied on less expressive power-of-two quanti-

zation scales. Lookup-table approaches seemed the most interesting, given that they can represent any value, compared to the limited number of steps representable through (sums of) power-of-two values.

### 2.3.4. Mixed Precision

Most works consider *homogeneous* quantization, which means that all weight or all activations across a model share the same bit-widths. Several recent works [28–31] explored the quantization of heterogeneous models, where each weight tensor in a layer can have its own bit-width, also commonly referred to as mixed-precision models. Assigning low bit-widths to layers that are less sensitive to quantization can increase compression and bandwidth efficiency even further. Moreover, a fine-grained trade-off can be selected between model accuracy and model compression by choosing the right configuration. The major challenge for mixed-precision quantization algorithms is to determine the bit-width configuration for an entire model. Several works [28–30] designed a sensitivity metric that represents the sensitivity to quantization for a given layer. A bit-width, proportional to this metric, is then assigned to that layer. Cai et al. [31] solved the selection of the right mixed-precision model through differential NAS.

Although mixed precision has many great advantages, it requires specialized hardware to support it. In this work, we did not consider using mixed precision; however, our method is suited to be used in combination with mixed precision, which can be explored in future work.

### 2.4. Quantizer Parameters

This section describes the different data types that are used in practice to represent quantizer parameters (Section 2.4.1) and how these parameter are estimated in the PTQ and QAT algorithms (Section 2.4.2).

### 2.4.1. Quantizer Parameter Data Types

The (simulated) data type of quantizer parameters such as scale factors and zero-points matters, since they influence the requirements of the target hardware. If, for example, $s$ would be a full-precision floating-point number during simulation, the accelerator would also need to support scaling operations in full-precision. Since floating-point operations in integer-based accelerators are usually not preferred, other solutions have been proposed in the literature for scale factors: dynamic fixed-point integers [13], which are 32 bit or 16 bit fixed-point integers plus a shift operation to enlarge their dynamic range, and powers-of-two values [12]. Zero-points are less critical and in practice represented through integers with the same bit-width as their integer data $\mathbf{q}$. Dynamic fixed-point integers are mostly used for scale factors since they closely resemble the dynamics and precision of floating-point numbers, which allows simulators to just represent them in floating-points. Power-of-two (POT) scale factors, according to Jain et al. [13], can be simulated with:

$$s_{POT} = 2^{\mathrm{ceil}(\log_2(s))} \tag{23}$$

$s$ is a full-precision scale factor; $s_{POT}$ is the power-of-two scale factor; ceil is a ceiling operator that rounds upwards. In this work, we preferred to use these power-of-two scale factors because they can be executed with a single bit-shift operation.

### 2.4.2. Quantizer Parameter Estimation

We found different techniques to estimate the quantizer parameters in both the PTQ and QAT frameworks. More traditional works estimate these parameters using simple data statistics, such as observed minimum and maximum data values [13,14,19], and use them as lower and upper bounds of the quantization grid, respectively. This approach is however sensitive to outliers and tends to overestimate the range, resulting in large quantization errors, especially for bit-widths smaller than eight. To overcome this issue,

many PTQ algorithms minimize some distance function $D$ between the full-precision data and the quantized data:

$$\arg\min_{s,z} \quad D\Big(\mathbf{x}, Q_f(\mathbf{x}, s, z)\Big) \qquad (24)$$

Different metrics for $D$ include the Kullback–Leibler divergence between both distributions [12,32] and the mean-squared error [18,33]. Modern QAT methods make quantizer parameters trainable and optimize them together with the weights [6,9,12,21,24,34,35]. Using the chain rule and the STE, the gradient w.r.t. the scale factor can be calculated. For the uniform symmetric quantizer, this will be:

$$\frac{\partial}{\partial s}\left[s\,\mathrm{clamp}\left(\mathrm{round}\left(\frac{\mathbf{x}}{s}\right), n, p\right)\right] = \begin{cases} \mathrm{round}\left(\frac{x}{s}\right) - \frac{x}{s} & \text{if} \quad n \leq \mathrm{round}\left(\frac{x}{s}\right) \leq p \\ n & \text{if} \quad \mathrm{round}\left(\frac{x}{s}\right) < n \\ p & \text{if} \quad \mathrm{round}\left(\frac{x}{s}\right) > p \end{cases} \qquad (25)$$

$n = -2^{b-1}$ and $p = 2^{b-1} - 1$ for signed and $n = 0$ and $p = 2^b - 1$ for unsigned data.

Some works define $s = 1$ and $z = 0$ [11,36,37], which completely eliminates scaling on the target device. The consequence, however, is that all layers have to learn quantized values that lie more or less within the same range, which works since these authors trained their models from scratch. Training from scratch with quantizers is, however, less preferable since it takes much longer to train and sometimes results in less accurate models.

As mentioned in Section 2.4.1, we preferred power-of-two scales. We optimized the scales through backpropagation as proposed by Jain et al. [12]. In the forward pass, the power-of-two scale is calculated with Equation (23). In the backward pass, the STE is used for the ceil function so $\frac{\partial\,\mathrm{ceil}(x)}{\partial x} = 1$, and $s_{log} = \log_2 s$ is treated as a training parameter rather than $s$ itself. The latter enables training in the $\log_2$ domain, which increases the numerical stability and provides better convergence. The gradient w.r.t. $s_{log}$ of Equation (23) is:

$$\frac{\partial}{\partial s_{log}}\left[2^{\mathrm{ceil}(s_{log})}\right] = 2^{\mathrm{ceil}(s_{log})}\ln 2 \qquad (26)$$

*2.5. Quantizing Other Layers*

Sections 2.1 and 2.2 discuss the internals of a fixed-point compute engine and how to simulate it, respectively. In this section, we discuss how other common layers, such as elementwise adders or concatenation layers, need to be handled in hardware in order to correctly deal with scale factors and zero-points:

**Max pooling**: This layer does not need special handling because it only has a single input branch and does not alter these input values numerically;

**Average pooling**: The average of multiple integers is not necessarily an integer, which requires inserting a *fake* quantization node at the output. In practice, average pooling is often executed through a depthwise convolution layer without bias and with weights $\mathbf{w} = 1/K^2\mathbf{J}$ where $K$ is the kernel size of the average pool layer and $\mathbf{J}$ is an all-ones tensor. This way, the convolution compute engine can be reused for average pool operations, and quantization simulation happens in the same way as for the convolution engine;

**Elementwise add**: If a scale or zero-point of both input branches of the adder do not match exactly, the quantization grids will be different and the addition will be incorrect. Krishnamoorthi et al. [14] placed a *fake* quantizer node at both input branches and one at the output. Their hardware implementation used a scale operation at both input branches and a round operator at the output. Jain et al. [12] had a slightly different approach. They forced the *fake* quantizer nodes at both input branches to share the same quantizer parameters, as depicted in Figure 5a by the dashed line. This way, the hardware implementation, shown in Figure 5b, only needs a requantization step at the output. In case a ReLU activation function directly follows the elementwise adder, which is for instance the case with ResNet, this function can be fused into the elementwise add operation through the clipping function from the requantization step;

**Concatenation**: The input branches of a concatenation function likely do not share the same scale or zero-point. Just as elementwise addition, the quantization parameters of the input branches need to be shared. This can be done in simulation by inserting *fake* quantization nodes on every input branch with shared parameters or by using a single *fake* quantization node at the output, which has the same effect (Figure 5c). The hardware implementation is shown in Figure 5d.

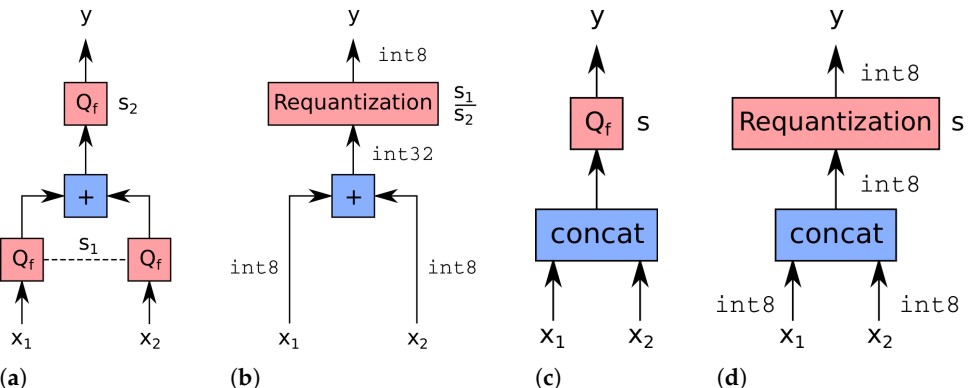

(a)                        (b)                        (c)                        (d)

**Figure 5.** *Fake* quantized elementwise add (**a**), its implementation in hardware (**b**), *fake* quantized concat (**c**), and its implementation in hardware (**d**). The figures assume a uniform symmetric quantization scheme (only scale factors) for the sake of simplicity.

*2.6. QAT Initialization*

The superior performance of QAT over PTQ is significantly influenced by proper initialization of the QAT process. It is known that initializing QAT with pretrained floating-point weights gives better results compared to training a quantized model from scratch [12–14]. The proper initialization of the quantizer parameters is however equally important, which is why QAT algorithms first perform PTQ to give the quantizer parameters a meaningful starting point. Nagel et al. [6] experimented with different initialization techniques for uniform QAT. They concluded that the mean-squared-error-based estimation of the quantization parameters is favorable compared to min-max-based estimation, though the advantage becomes small towards the end of the training. Other techniques to nudge the weights or biases prior to training such as AdaRound [38] or bias-correction [22] seem not to be beneficial, with the exception of Cross-Layer Equalization (CLE) [22] which, according to their experiments, is required in order to successfully train MobileNets with 4 bit weights.

In addition to PTQ initialization, QAT-trained models with higher bit-widths are also used to initialize QAT trainings with lower bit-widths, in which case the bit-width is progressively reduced in multiple training stages. Commonly known as *progressive* quantization, this strategy has proven to work well and gives a small boost to the final accuracy. Progressive quantization can also mean that parts of the network are quantized progressively rather that the whole network at once. Zhuang et al. [7] proposed to first quantize the weights and leave the activations in full-precision, followed by quantizing both the weights and activations in a second stage. Park et al. [9] exploited this idea even further and proposed PROFIT, a technique to identify layers that cause more instability during QAT than others in MobileNets. By freezing the most unstable layers (mainly depthwise convolutions) in a consecutive training stage, that stage would be trained more smoothly, resulting in better performance. Progressive freezing, by Park et al. [9], was performed in three stages: (1) all layers were trained; (2) 2/3 of the layers were trained; finally, (3) 1/3 of the layers were trained. Each training stage was initialized with the weights from the previous training stage.

We applied CLE for all our MobileNetV2 experiments and used PROFIT in some experiments to further close the gap between quantized and full-precision models.

## 3. Materials and Methods

As discussed in our related work, most methods rely on high-precision scaling to cope with the different dynamic data ranges, present in full-precision neural networks. High-precision scaling requires additional high-precision multipliers, which require larger silicon area and more energy and, in case FPGAs are used, leads to more expensive platforms. We propose a novel method that avoids high-precision scales in fixed-point compute engines by using power-of-two scales, which can be applied through a single bit-shift. Jain et al. [12] already proved that quantized models with 8 bit weights and power-of-two quantization scales can achieve full-precision accuracy or even better. Although much harder, we prove that we can achieve full-precision accuracy with 4 bit weights and power-of-two scales. Our method was inspired by the superior performance of nonuniform quantizers over uniform quantizers, but does not require complex hardware.

The difference between a typical compute engine and our proposed engine is depicted in Figure 1a,b, respectively. Our compute engine relies on bit-shift-based scaling with a single scale per layer, and we propose to use a lookup-table to translate 4 bit weights into 8 bit weights for internal computation. The lookup-table allows the compute engine to have nonuniform properties, boosting its expressiveness significantly. Since the LUT can hold any number representable through eight bits, different value combinations can be chosen to match the underlying weight distribution within a specific convolution layer. A different set of LUT values is used for every layer to best match the layer-specific data distribution. Note that a single LUT can be shared among multiple parallel compute engines, allowing an additional hardware simplification.

Section 3.1 presents our quantization scheme, the design changes to the compute engine, and the *fake* quantizer that we propose to simulate our design. Section 3.2 discusses the proposed optimization algorithm to estimate the 8 bit lookup-table values for each layer, and Section 3.3 describes the initialization method of both lookup-table values and power-of-two scales, used prior to training.

### 3.1. Lookup-Table Quantizer

Our quantization scheme can be expressed by Equation (27), where a floating-point vector $\mathbf{x}$ is approximated by a power-of-two scale factor $s$, multiplied by the output of LUT function $L$. $L$ simply returns the integer values from LUT content $\mathbf{v} \in [-2^{b-1} ... 2^{b-1} - 1]^K$, which correspond to quantized values $\mathbf{q}$ (the lookup addresses). Here, $K$ is the number of values in the LUT and $b$ is the number of bits to represent them.

$$\mathbf{x} \approx sL(\mathbf{q}, \mathbf{v}) \qquad s.t. \quad s = \frac{2^l}{2^{b-1}} \tag{27}$$

If, for example, $\mathbf{q}$ contains 4 bit addresses, the LUT will contain $K = 16$ values, each with b bits (typical $b = 8$). Because $\mathbf{v}$ is a set of discrete b bit values rather than floating-point values, we used the power-of-two scale factor $s$ to increase the dynamic range of the scheme, where $l \in \mathbb{Z}$ is the amount of left-shifting. Figure 6 illustrates this scheme graphically.

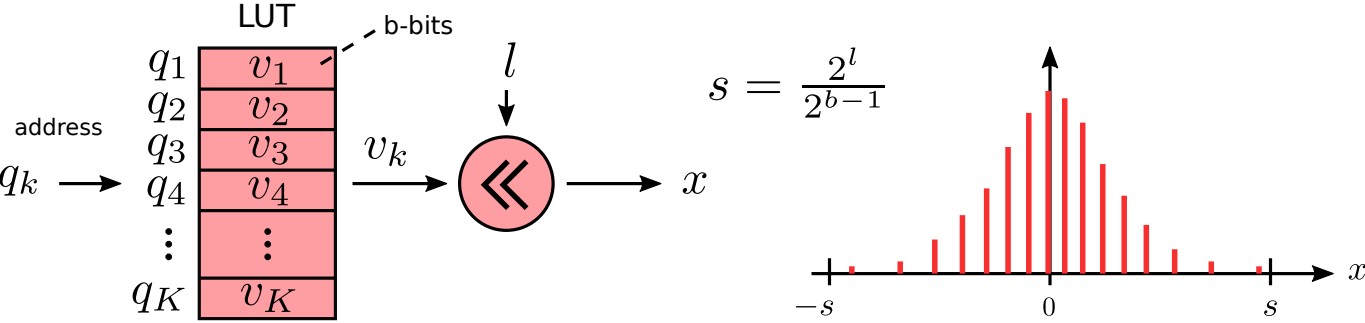

**Figure 6.** Our quantization scheme, which calculates a value $x$ by translating a lookup-address $q_k$ into a b bit value $v_k$ and bit-shifts the results by $l$ places.

Although this quantization scheme is symmetric (since it lacks a zero-point), it has asymmetric properties since the LUT values can be distributed in an asymmetric way if needed. If Equation (27) is used to quantize the weights within a b bit multiply-accumulate engine with uniform symmetric activations, we obtain:

$$s_y y_q = s_o o_q + \sum^N s_a \mathbf{a}_q \odot s_w L(\mathbf{w}_q, \mathbf{v}) \tag{28}$$

$$y_q = S\left(o_q + \sum^N \mathbf{a}_q \odot L(\mathbf{w}_q, \mathbf{v})\right) \qquad s.t. \quad S = \frac{s_a s_w}{s_y} \tag{29}$$

Here, $\mathbf{a}_q$ and $L(\mathbf{w}_q, \mathbf{v})$ represent activation and weight vectors, respectively. Scale factors $s_y$, $s_a$, and $s_w$ are all power-of-two scale factors that can be merged into a single power-of-two scale factor $S$. The engine in Equation (29) only requires N LUT operations, N b bit multiply-accumulate operations, a bias add operation, and a single bit-shift operation to calculate the output.

A *fake* quantization node of the quantization scheme from Equation (27) is presented in Equation (30):

$$Q_f(\mathbf{x}, s, \mathbf{v}) = s\,\mathrm{proj}\left(\frac{\mathbf{x}}{s}, \mathbf{v}\right) \tag{30}$$

Here, projection function $\mathrm{proj}(x, \mathbf{v})$ maps input $x$ onto the value in $\mathbf{v}$ that lies closest to $x$ and $s$ is the power-of-two scale factor.

In the backward pass, the Straight-Through Estimator (STE) principle is applied, which means that the projection function behaves as an identity function, as shown by Equation (31):

$$\frac{\partial\,\mathrm{proj}(x, \mathbf{v})}{\partial x} = 1 \tag{31}$$

Each weight tensor within a neural network has its own $s$ and $\mathbf{v}$ parameter set, which needs to be optimized. First, we used a PTQ algorithm to determine the initial values of both $s$ and $\mathbf{v}$. Second, during QAT, $s$ remains constant, and $\mathbf{v}$ is further optimized together with the weights.

Section 3.2 explains the algorithm to optimize $\mathbf{v}$ in both the PTQ and QAT stages, and Section 3.3 describes the steps taken during our PTQ stage, which initializes both $s$ and $\mathbf{v}$.

*3.2. Optimizing LUT Values* $\mathbf{v}$

Our algorithm for optimizing $\mathbf{v}$ is based on minimizing the quantization error between the floating-point data $\mathbf{x} = [x_1, x_2, \ldots, x_N]$ and its quantized result $Q_f(\mathbf{x}, s, \mathbf{v})$ by optimizing the LUT vector $\mathbf{v}$, as shown by Equation (32):

$$\mathbf{v} \leftarrow \underset{\mathbf{v}}{\mathrm{argmin}} \left\| \mathbf{x} - Q_f(\mathbf{x}, s, \mathbf{v}) \right\|_2^2 \tag{32}$$

We solve Equation (32) through an iterative algorithm, where a single iteration is described by Algorithm 1. First, the algorithm selects a subset of values $\mathbf{z}$ from scaled input data $\mathbf{y}$ that contain only the values of $\mathbf{y}$ that are nearest to LUT value $v_k$. Second, the mean value of $\mathbf{z}$ is calculated and used as the new LUT value after saturating it within interval $[-2^{b-1}, 2^{b-1} - 1]$ through a clamp operator, which simulates the upper and the lower bounds of the LUT. Note that this is the same as applying a single step of a K-means clustering algorithm, with the addition of the clamp operator. These steps are repeated for all values in $\mathbf{v}$.

The range limitation through the *clamp* operator is controlled by scale factor $s$. Figure 7 illustrates the outcome of Algorithm 1 when iteratively applied for two different values of $s$, on one of the weight tensors from MobileNetV2. A smaller value of $s$ results in more densely positioned quantization levels and excludes more outliers.

$\mathbf{v}$ is limited in range, but the updates to $\mathbf{v}$ are still performed in full-precision. To ensure that $\mathbf{v}$ can be represented exactly through b bit integers, rounding is applied after

some time of training when the values in **v** tend to be stabilized. Once rounded, the optimization process of **v** is stopped, and training continues with fixed LUT values. This can be seen in Algorithm 2, which summarizes the full algorithm that quantizes **x** and only optimizes **v** when Boolean flag *enable_optimization* is true. During QAT, Algorithm 2 is called once every forward pass.

We kept the update process of **v** in full-precision for the following two reasons: (1) direct rounding during optimization causes the optimization process to stall early when the updates become small, and (2) constant integer updates cause instability to the training process.

The criterion that determines when to set *enable_optimization* to false is explained in Section 3.2.1.

---

**Algorithm 1:** Optimize the LUT values.

**Input:** Floating-point data $\mathbf{x} = [x_1, x_2, \ldots x_N]$, scale factor $s$, and LUT values
  $\mathbf{v} = [v_1, v_2, \ldots v_K]$ where $v_1 < v_2 < \cdots < v_k$
**Output:** Updated LUT values $\mathbf{v} = [v_1, v_2, \ldots v_K]$
**Procedure:**
$\mathbf{y} = \mathbf{x}/s$
**for** $k = 1$ *to* $K$ **do**

$\quad t^- \leftarrow \begin{cases} -\infty & \text{if } k = 1 \\ v_k - \frac{v_k - v_{k-1}}{2} & \text{otherwise} \end{cases}$

$\quad t^+ \leftarrow \begin{cases} \infty & \text{if } k = N \\ v_k + \frac{v_{k+1} - v_k}{2} & \text{otherwise} \end{cases}$

$\quad \mathbf{z} \leftarrow \{y_i \in \mathbf{y} | t^- < y_i \leq t^+\}$
$\quad \bar{z} \leftarrow \text{mean}(\mathbf{z})$
$\quad v_k \leftarrow \text{clamp}\left(\bar{z}, -2^{b-1}, 2^{b-1} - 1\right)$

**end**

---

**Algorithm 2:** Quantize data during training/testing and optimize LUT values **v**.

**Input:** Floating-point data $\mathbf{x} = [x_1, x_2, \ldots x_N]$, scale factor $s$, and LUT values
  $\mathbf{v} = [v_1, v_2, \ldots v_K]$
**Output:** *fake* quantized data $\mathbf{q} = [q_1, q_2, \ldots q_N]$ and updated LUT values **v**
**Procedure:** // Run once in every forward pass

$\quad \mathbf{q} \leftarrow s \, \text{proj}(\frac{\mathbf{x}}{s}, \mathbf{v})$ // Apply fake quantization
$\quad$ **if** *enable_optimization* **then**
$\quad\quad \mathbf{v} \leftarrow$ Algorithm 1$(\mathbf{x}, s, \mathbf{v})$
$\quad$ **else**
$\quad\quad \mathbf{v} \leftarrow \text{round}(\mathbf{v})$ // round **v** and stop optimizing
$\quad$ **end**

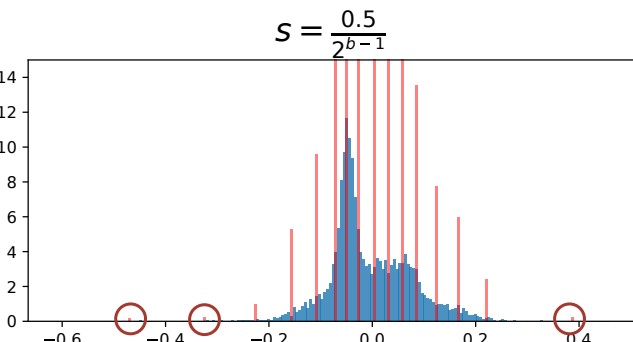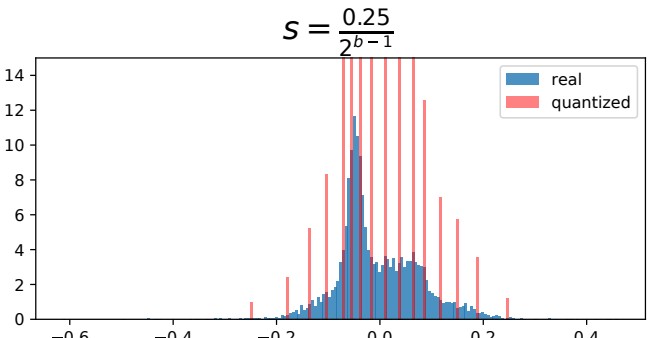

**Figure 7.** Real and quantized (16 bins) distributions of a weight tensor from MobileNetV2, generated using Algorithm 1. On the left, a few quantization bins are assigned to outliers due to the larger scale factor *s*. On the right, a smaller *s* results in a more compact quantized distribution because of the clamp operator.

### 3.2.1. Stopping Optimization of the LUT Values

Each weight layer has its own LUT vector **v** within a model. During training, when a LUT vector **v** is stabilized, it is rounded and its optimization procedure is stopped, which we call *freezing* (setting *enable_optimization* to false). We considered a vector stabilized when the following criteria are met:

$$\text{round}(\mathbf{v}) = \text{round}(\mathbf{v}_{smooth}) \tag{33}$$

where $\mathbf{v}_{smooth}$ is a smoothed version of **v** calculated by an EMA filter with a high decay (0.999). We start checking all LUT vectors after the first 1000 training iterations every 50 cycles. We only froze one LUT vector at a time. In case the criteria are met for multiple **v**, the vector with the smallest round error $\|\mathbf{v} - \text{round}(\mathbf{v})\|_2$ is preferred.

### 3.3. Initialization

Both scale factor *s* and LUT values **v** are initialized prior to training using our PTQ as the initialization method. Our initialization method, summarized in Algorithm 3, starts with uniformly distributed LUT values **v** and a power-of-two scale factor determined from the maximum value of **x**. It then optimizes **v** through Algorithm 2 for that specific scale factor followed by calculating the mean-squared error $\epsilon$ between quantized data **q** and **x**. This is repeated for five smaller power-of-two values of *s*. Optimal scale factor $\hat{s}$ and LUT vector $\hat{\mathbf{v}}$ that produced the lowest $\epsilon$ are selected as the final initialization values for training. Note that our PTQ does not yet round the values in **v**.

Algorithm 3 is applied for all weight tensors. Batch normalization layers are folded prior to quantization as discussed in Section 2.2.2. We did not quantize bias parameters in our experiments since we assumed 32 bit integer bias parameters and accumulators.

### 3.4. Quantizing Feature-Map Data

For quantizing feature-map data or activations, 8 bit quantization with power-of-two quantization scales was used. This was implemented through the symmetric uniform quantizer from Jain et al. [12] with learnable power-of-two scales. Their hyperparameter settings and scale freezing technique for optimizing the scales were adopted as well. Note that we deliberately did not choose to use asymmetric quantizers for feature-map data. Although this is possible, it introduces additional zero-points that need optimization, which complicates the training process. In addition, we empirically discovered that the added value is virtually nothing at 8 bit compared to a mix of signed and unsigned symmetric quantizers at the right locations. Throughout the model, quantizers are inserted at the inputs, activation functions, and other nonparametric layers such as elementwise adders, concatenations and average pooling layers. This is done to achieve correct integer-only inference simulation, as discussed in Section 2.5.

---

**Algorithm 3:** Initialization of LUT values **v** and scale factor $s$.

---

> **Input:** Floating-point data $\mathbf{x} = [x_1, x_2, \ldots x_N]$
> **Output:** Optimal scale factor $\hat{s}$ and LUT values $\hat{\mathbf{v}} = [v_1, v_2, \ldots v_K]$
> **Procedure:**
> $\hat{\epsilon} \leftarrow \infty$
> $\hat{s} \leftarrow s \leftarrow \frac{2^l}{2^{b-1}} \quad s.t. \quad l = \text{ceil}(\log_2 \max |\mathbf{x}|)$
> **for** $i = 1$ *to* 5 **do**
> > $\mathbf{v} = [v_1, v_2, \ldots v_K]$ with uniformly distributed values in range $[-2^{b-1}, 2^{b-1} - 1]$
> > **for** $j = 1$ *to* 100 **do**
> > > $\mathbf{q}, \mathbf{v} \leftarrow$ Algorithm 2$(\mathbf{x}, s, \mathbf{v})$    // Optimization enabled
> >
> > **end**
> > $\epsilon \leftarrow \|\mathbf{x} - \mathbf{q}\|_2^2$
> > **if** $\epsilon < \hat{\epsilon}$ **then**
> > > $\hat{\epsilon} \leftarrow \epsilon \quad \hat{s} \leftarrow s \quad \hat{\mathbf{v}} \leftarrow \mathbf{v}$
> >
> > **end**
> > $s \leftarrow s/2$
>
> **end**

---

## 4. Results

To evaluate the performance of our LUT-based method, we conducted several experiments on CNN-based image classifier models and an object detection model. Section 4.1 compares the PTQ and QAT performance of our proposed method against other uniform quantizers with both full-precision and power-of-two quantization scales in both per-layer and per-channel mode. Section 4.2 compares our results against results from other state-of-the-art works for five different image classifier models. Section 4.3 presents the results of our method, tested on an object detector model. Section 4.4 presents a small study on the added value of optimizing the LUT values during QAT. Section 4.5 discusses additional tools to further improve the accuracy of MobileNetV2. Finally, in Section 4.6, we present the simulated inference speed of a few CNN models with both 8 and 4 bit weights on an FPGA platform.

### 4.1. Comparison to Different Quantizers

In this first set of experiments, we compared our proposed nonuniform weight quantization method against uniform quantization methods with the POT scale and float scale in both the per-layer and per-channel configurations. Table 1 lists the Top-1 ImageNet accuracies of quantized ResNet18 [39] and MobileNetV2 [40] with 4 bit weights and 8 bit activations. We included both QAT's and PTQ's results, where the PTQ models were used to initialize the QAT models. Note that all these results were produced using the same framework for a fair comparison.

Table 1 shows that our method achieved the highest scores for both models, even slightly outperforming per-channel quantized models with full-precision scales. The advantage of the lookup-table clearly made the difference: our method can be used as a worthy alternative for uniform quantizers with both per-layer and per-channel quantization with floating-point scales. We also noticed that uniform quantization with POT scales in both per-layer and per-channel quantization did not completely recover the baseline accuracy of ResNet18, while our method did with a safe margin of +0.35%, which proves the superiority of our method for POT scales.

It is known that the accuracy of QAT models can exceed the accuracy of their full-precision counterparts [6,12,14], which can also be seen with our method. This phenomenon can be explained by the additional training cycles of QAT and the regularization effect of the quantization noise during QAT.

**Table 1.** ImageNet Top-1 results using different quantizers with 4 bit weights and 8 bit activations. The first number represents the relative accuracy or difference between the quantized model and the full-precision model, while the second number in parenthesis represents the absolute accuracy.

| Model (FP32 Accuracy) | ResNet18 (69.76) | | MobileNetV2 (71.88) | |
|---|---|---|---|---|
| | PTQ | QAT | PTQ | QAT |
| Float scale per-layer | −50.18 (19.58) | +0.08 (69.84) | −67.11 (4.77) | −1.51 (70.37) |
| Float scale per-channel | −15.85 (53.91) | +0.23 (69.99) | −64.96 (6.92) | −0.97 (70.91) |
| POT scale per-layer | −60.14 (9.62) | −0.27 (69.49) | −71.05 (0.83) | −1.73 (70.15) |
| POT scale per-channel | −31.75 (38.01) | −0.08 (69.68) | −69.87 (6.34) | −1.66 (70.22) |
| POT scale per-layer + LUT (ours) | −12.51 (57.25) | **+0.35 (70.11)** | −55.95 (15.93) | −**0.92 (70.96)** |

Although per-channel quantization with POT-scales was never attempted by Jain et al. [12], their per-layer results were consistent with ours. Per-channel quantization with POT scales has, to the best of our knowledge, never been tried in the literature before. Our procedure for uniform per-channel quantization with POT scales adopted the gradual freezing method of Jain et al. [12] to freeze the POT scales once stabilized, but applied it on a per-channel level. Instead of allowing a single POT scale to be frozen every 50 training iterations, we allowed up to 400 per-channel scales to be frozen, using the same stabilization criteria from Jain et al. [12].

For MobileNetV2, however, no quantizer succeeded in recovering the full-precision accuracy, because it is very sensitive to quantization due to the depthwise separable convolutions. Although CLE [22] was applied prior to quantization to improve MobileNetV2's QAT accuracy in all our experiments, as suggested by Nagel et al. [6], a drop in accuracy of approximately one percent still existed. In Section 4.5, we further analyze the effect of CLE on POT-scale-based quantizers in more detail, and we discuss what can be done in addition to further improve the results of MobileNetV2.

Both models were trained for 20 epochs on ImageNet, with standard preprocessing, data augmentation, and batch sizes of 256 and 128 for ResNet18 and MobileNetV2, respectively. We used the Adam optimizer to optimize both weights and quantization scales with its default parameters. For the weights, learning rates of $1 \times 10^{-5}$ and $3 \times 10^{-5}$ were used for ResNet18 and MobileNetV2, respectively, with a cosine learning rate decay that reduced the learning rate to zero at the end of the training. The quantization scales had a learning rate of $1 \times 10^{-2}$ that quickly decayed in a step-based fashion, as suggested by Jain et al. [12] for their POT scale quantizer. We discovered that this learning rate schedule worked better for both POT-scale and float-scale models compared to the same learning rate schedule as used for the weights. Note that other best practices to train our POT-scale models, including gradual freezing of the scales once stabilized, were also adopted from Jain et al. [12], for our experiments that optimize POT scales.

The PTQ methods for the uniform models used mean-squared error estimation to initialize both weight and activation scales, as suggested by Nagel et al. [6]. Note that all experiments used symmetric quantizers for the activations.

*4.2. Comparison to the State-of-the-Art*

In this section, we compare our method against results from other state-of-the-art methods for a number of different models. The selection criteria used to select works to compare to were twofold: first, mainly QAT methods and, second, works that reported results with 4 bit weights and 8 bit activations. Table 2 presents the Top-1 ImageNet accuracies of ResNet18 [39], ResNet50 [39], MobileNetV2 [40], VGG16 [41] without batch normalization, and InceptionV3 [42]. The accuracies of the full-precision models among the related works slightly differed and are therefore listed separately (32/32) in addition to the accuracies of the quantized models with 4 bit weights and 8 bit activations (4/8). In addition to the results of our method (*Ours LUT*), we also present our own results of uniform quantization with POT scales (*Ours uniform*), to highlight the added value of our LUT.

Results from the other works were directly taken from their papers. AB [18], QAT1 [14], and QAT2 [6] are all QAT methods that use per-channel quantization with full-precision quantization scales, where AB and QAT2 also experimented with per-layer quantization. TQT [12] is a QAT method that uses POT scales with per-layer quantization, and PWLQ [43] is a PTQ method that uses multiple uniform quantizers to obtain nonuniform quantization.

We can conclude that all models using our method achieved and even exceeded full-precision model accuracy with a safe margin, with the exception of MobileNetV2. Although the latter model was not fully recovered, we still achieved the highest accuracy compared to other methods. Nagel et al. [6] achieved a slightly higher accuracy on ResNet50 and InceptionV3 using per-channel quantization with float scales, but comparable to ours.

For ResNet50, VGG16, and InceptionV3, we used a learning rate $3 \times 10^{-6}$ and their standard respective preprocessing and data augmentation. All other hyperparameters were as described in the previous Section 4.1. InceptionV3 was fine-tuned without auxiliary loss, as suggested by Jaint et al. [12].

In Table 3, we also present the ImageNet results of uniform quantized models with power-of-two scales and 8 bit weights instead of 4 bit weights, to further put our 4/8 results into perspective. Overall, these results were slightly higher compared to our results from Table 2, which was however to be expected. Similar training conditions were used as with our 4/8 experiments.

**Table 2.** Top-1 ImageNet accuracies of different works. The first value represents the relative accuracy or difference between the quantized model and the full-precision model, while the second number in parenthesis represents the absolute accuracy. Columns Per-Ch and POT scale indicate whether per-channel quantization or POT scales were used, respectively.

| Method | w/a | Per-Ch | POT Scale | ResNet18 | ResNet50 | MobileNetV2 | VGG16 | InceptionV3 |
|---|---|---|---|---|---|---|---|---|
| AB [18] | 32/32 | | | / | 75.20 | / | / | / |
| | 4/8 | no | no | / | −1.40 (73.80) | / | / | / |
| | 4/8 | yes | no | / | −0.90 (74.30) | / | / | / |
| PWLQ [43] | 32/32 | | | / | 76.13 | 71.88 | / | 77.49 |
| | 4/8 | yes | no | / | −0.51 (75.62) | −2.68 (69.22) | / | −1.04 (76.45) |
| TQT [12] | 32/32 | | | / | 75.20 | / | 70.90 | 78.00 |
| | 4/8 | no | yes | / | −0.80 (74.40) | / | **+0.60** (71.50) | −1.60 (76.4) |
| QAT1 [14] | 32/32 | | | / | 75.20 | 71.90 | / | 78.00 |
| | 4/8 | yes | no | / | −2.00 (73.2) | −9.90 (62.00) | / | −2.00 (76.00) |
| QAT2 [6] | 32/32 | | | 69.76 | 76.07 | 71.72 | / | 77.40 |
| | 4/8 | no | no | +0.08 (69.76) | −0.18 (75.89) | −1.55 (70.17) | / | +0.44 (77.84) |
| | 4/8 | yes | no | +0.33 (70.01) | **+0.45 (76.52)** | −1.24 (70.48) | / | **+0.72 (78.12)** |
| Ours uniform | 32/32 | | | 69.76 | 76.13 | 71.88 | 71.59 | 77.34 |
| | 4/8 | no | yes | −0.27 (69.49) | −0.11 (76.02) | −1.73 (70.15) | +0.09 (71.68) | +0.23 (77.57) |
| Ours LUT | 4/8 | no | yes | **+0.35 (70.11)** | +0.18 (76.31) | −**0.92 (70.96)** | +0.28 (**71.87**) | +0.39 (77.73) |

**Table 3.** Top-1 ImageNet accuracies of uniform quantized models with 8 bit weights and 8 bit activations using POT scales.

| w/a | ResNet18 | ResNet50 | MobileNetV2 | VGG16 | InceptionV3 |
|---|---|---|---|---|---|
| 32/32 | 69.76 | 76.13 | 71.88 | 71.59 | 77.34 |
| 8/8 | +0.74 (70.50) | +0.54 (76.67) | +0.00 (71.88) | +0.37 (71.96) | +0.99 (78.33) |

*4.3. Object Detection*

We also tested our method on the Tiny YOLOV2 [44] object detector. Table 4 presents the mean-average-precision results of Tiny YOLOV2 on the MS COCO dataset with 4 bit weights and 8 bit activations. The model was quantized with per-layer POT scales, both with and without our LUT method. We can conclude that also for this object detector, our method improved the final QAT result compared to uniform quantization. We trained the

model for approximately 27 epochs with a learning rate of $1 \times 10^{-6}$ and a batch size of 32. An input resolution of $416 \times 416$ pixels was used with the standard data augmentation of YOLO, leaving out model rescaling during QAT.

**Table 4.** Detection results on MS COCO in mean average precision (IoU 0.5) with 4 bit weights and 8 bit activations.

| Model (FP32 Accuracy) | Tiny YOLOV2 (22.51) | |
|---|---|---|
| | PTQ | QAT |
| POT scale per-layer | −17.50 (5.01) | +0.1 (22.61) |
| POT scale per-layer + LUT (ours) | −20.80 (1.71) | **+0.94 (23.45)** |

*4.4. Effect of Optimizing LUT Values during QAT*

To study the added value of the optimization (Section 3.2) and progressive freezing (Section 3.2.1) of the lookup-table values during QAT, we conducted a small ablation study, presented by Table 5. The first two columns present the Top-1 accuracy of ResNet18 and MobileNetV2, while Columns 3 and 4 indicate whether optimization of the LUTs and progressive freezing were enabled, respectively. We can conclude that enabling the optimization of all LUTs during the whole QAT procedure without progressive freezing was not beneficial; it even resulted in a reduced accuracy compared to fixed LUT values during QAT. However, enabling optimization during the initial phase of QAT by progressively stopping and rounding the LUT values when stabilized resulted in the best final accuracy.

The results of the first row were obtained by applying our PTQ algorithm first, followed by rounding the lookup-table values and finally running QAT with constant lookup-table values. For the results of the second row, the LUT values were never rounded, and their optimization was enabled until the end of the training.

**Table 5.** Study on the added value to the Top-1 accuracy of optimizing the LUT values during QAT and the effect of progressive freezing. Experiments were conducted with 4 bit weights and 8 bit activations.

| ResNet18 (69.76) | MobileNetV2 (71.88) | Optimize LUT during QAT | Progressive Freezing |
|---|---|---|---|
| +0.31 (70.07) | −1.07 (70.81) | | |
| −0.11 (69.65) | −2.39 (69.49) | ✓ | |
| **+0.35 (70.11)** | **−0.92 (70.96)** | ✓ | ✓ |

*4.5. Improving Results for MobileNetV2*

This section presents an ablation study on the effect of the CLE [22] and PROFIT [9] methods on MobileNetV2, quantized with our method. Table 6 indicates that CLE increased the QAT accuracy by 0.24%. Since CLE was applied prior to quantization and did not need additional training, we applied it to all our other MobileNetV2 experiments.

As explained in Section 2.6, PROFIT is a technique to improve models with depthwise convolutions, such as MobileNets. In contrast to CLE, PROFIT requires two additional training stages on top of the standard QAT procedure. Our first PROFIT stage trained with 1/3 of the layer weights frozen, while the second stage trained with 2/3 of the layer weights frozen, where both stages trained for 20 epochs each (60 epochs in total, including initial QAT). With PROFIT, we could further increase the accuracy by another 0.41% to arrive at a final relative accuracy of −0.51%. During each PROFIT stage, we kept our quantizer parameters $s$ and $\mathbf{v}$ frozen and used the same learning rate and learning rate schedule as used in the initial QAT, but with a ramp-up schedule at the start.

**Table 6.** Ablation study of the CLE [22] and PROFIT [9] methods on quantized MobileNetV2 with our quantization method. PROFIT Stage 1 is a second round of training where 1/3 of the layers have frozen weights, and PROFIT Stage 2 is a third round of training where 2/3 of the layers have frozen weights.

| MobileNetV2 (71.88) | CLE | PROFIT Stage 1 | PROFIT Stage 2 |
|---|:---:|:---:|:---:|
| −1.16 (70.72) | | | |
| −0.92 (70.96) | ✓ | | |
| −0.67 (71.21) | ✓ | ✓ | |
| **−0.51 (71.37)** | ✓ | ✓ | ✓ |

*4.6. Speed Improvements*

As already mentioned in the Introduction, reducing the number of bits per weight from eight to four not only reduces the storage cost by 50%, it also increases the memory-bandwidth efficiency, resulting in faster inference. Table 7 presents simulation results in frames-per-second (FPS) of models with 8 bit weights and 4 bit weights on an Intel Arria 10 FPGA with 1024 MAC cores. For models with large weight tensors such as VGG16, performance boosts of up to 55% can be achieved, while for more sparse models such as MobileNetV2, the speed improvement was rather small. Simulations were performed using the *nearbAi* estimator software https://www.easics.com/deep-learning-fpga (accessed on 16 November 2021), which models the inference behavior of the *nearbAi* neural network accelerator IP core. A 32 bit-wide memory bus was used for the storage of weights and feature-maps in external RAM with a clock frequency of 230 MHz. A clock frequency of 200 MHz was used for the compute engine.

**Table 7.** Simulated speeds of models with 8 bit weights and 4 bit weights for an Intel Arria 10 FPGA with 1024 multiply-accumulate DSP cores and a 32 bit-wide bus to external RAM memory. Values are in Frames Per Second (FPS).

| | VGG16 | ResNet50 | MobileNetV2 |
|---|:---:|:---:|:---:|
| 8/8 | 4.2 | 14.9 | 37.2 |
| 4/8 | (+55%) 6.5 | (+28%) 19.1 | (+9%) 40.5 |

## 5. Conclusions

In this work, we highlighted the benefits of power-of-two quantization scales in quantized CNNs: scaling can be applied through a bitwise shift and does not require expensive high-precision multipliers. We showed that, with 4 bit weights, however, these quantized models are typically less accurate compared to models with high-precision scales, which is mainly caused by the less expressive power-of-two scale. Most of the models with bit-shift scales therefore did not recover the full-precision model accuracy. To solve this problem, we proposed to add a lookup-table to the compute engine that could translate 4 bit weight addresses into 8 bit nonuniformly distributed weights for internal computation, which greatly improved the overall expressiveness of the compute engine. We also noted that a single lookup-table could be shared among multiple parallel compute cores, further simplifying the overall design.

Through experiments, we proved that our method is capable of achieving the same accuracy as per-channel quantized models with full-precision scales, while only using a single power-of-two scale for a whole layer and a low-precision lookup-table. This allowed us to recover or even exceed the full-precision accuracy of various CNN models and achieve state-of-the-art results for the hard-to-quantize MobileNetV2. Reducing the number of bits per weight from eight to four also allowed us to reduce the model size by 50% and improved the bandwidth efficiency, resulting in increased inference speeds on an FPGA platform of up to 55%.

In future work, our method can be used easily in mixed-precision computation, where each weight tensor in a layer can have its own bit-width, to achieve a more fine-grained trade-off between model compression and model accuracy. Models with a mix of 2 bit, 4 bit, and 8 bit weights could be constructed and executed on our proposed compute engine without any modification. A 2 bit layer could be configured by simply using only four entries of the LUT, while an 8 bit layer could be configured through a simple bypass of the LUT.

**Author Contributions:** Methodology, M.V.; software, M.V.; supervision, T.G.; validation, K.V.B.; writing—original draft, M.V.; writing—review and editing, M.V., K.V.B., and T.G. All authors have read and agreed to the published version of the manuscript.

**Funding:** This work was funded by VLAIO and EASICS https://www.easics.com (accessed on 16 November 2021) through the Start to Deep Learn TETRA project and FWO via the OmniDrone SBO project.

**Data Availability Statement:** The following publicly available datasets were used in this research: the ImageNet dataset can be found here https://www.image-net.org/download.php and the MS COCO dataset can be found here https://cocodataset.org/#download (both accessed on 16 November 2021). The code used to conduct this research can be found here https://gitlab.com/EAVISE/lut-model-quantization (accessed on 16 November 2021).

**Conflicts of Interest:** The authors declare no conflict of interest.

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
