# Peer review of "Integer-Only CNNs with 4 Bit Weights and Bit-Shift Quantization Scales at Full-Precision Accuracy"

_electronics, doi:10.3390/electronics10222823_

Round 1
Reviewer 1 Report
The manuscript shows a hardware architecture for implementing integer-based convolutional neural networks.
The paper includes sufficient related works, the methods are clearly presented, and the results are adequately described.
However, I have an amendment.
Since the design has been implemented on an Arria FPGA it would be good to add a table with the implementation details. Some data about the maximum clock frequency, power dissipation, required hardware resources.., throughput,... The more details, the better.
Author Response
Thank you for taking the time to read and evaluate our article, your feedback is much appreciated.
We have added some details about clock frequencies to Section 4.6. However, since the speed results in Section 4.6 are simulation results for a specific FPGA platform rather than results from an actual implementation, we are not able to provide further details on power dissipation and required resources. An actual implementation of our method is still work in progress and would require more time to accomplish.
Reviewer 2 Report
The paper has provided the details of the approach applied. The results are well briefed with the comparision between ResNet18 and MobileNetV2 also with respect to optimising the weights and Quantization Scales with default parameters.
The work can be published in the present form.
Author Response
Thank you for taking the time to read and evaluate our article, your feedback was much appreciated.
Reviewer 3 Report
The author proposes a non-uniform quantization method with 4-bit weights and 8-bit activations. The idea is interesting. However, there are still some problems in this article. Therefore, it needs to be revised before publication. And my comments for the authors are as follows:
- The author utilizes LUT to represent non-uniform weights. However, it needs additional operations to access LUT during calculation, and the author needs to evaluate the overhead.
- Section 4.1, Table1, the author doesn’t give an adequate explanation why its model accuracy is higher than the full-precision model. In addition, models were trained for 20 epochs are not enough to get the best accuracy.
- Section 4.1, Table2, the author only analysis the advantage of ResNet18 and MobileNet, the disadvantage of other models also need to be analyzed.
- Section 4.2, Table2, it is necessary to compare with the non-uniform algorithm work, and 8/8 w/a should also be added to show the advantage on author's 4/8 work. In addition, most of the comparative literature is arXiv preprint, which is not appropriate.
